# Primary Chemoradiotherapy Treatment (PCRT) for HER2+ and Triple Negative Breast Cancer Patients: A Feasible Combination

**DOI:** 10.3390/cancers14184531

**Published:** 2022-09-19

**Authors:** Raquel Ciérvide, Ángel Montero, Eduardo García-Rico, Mariola García-Aranda, Mercedes Herrero, Jessica Skaarup, Leticia Benassi, Maria José Barrera, Estela Vega, Beatriz Rojas, Raquel Bratos, Ana Luna, Manuela Parras, María López, Ana Delgado, Paloma Quevedo, Silvia Castilla, Margarita Feyjoo, Ana Higueras, Mario Prieto, Ana Suarez-Gauthier, Lina Garcia-Cañamaque, Nieves Escolán, Beatriz Álvarez, Xin Chen, Rosa Alonso, Mercedes López, Ovidio Hernando, Jeannette Valero, Emilio Sánchez, Eva Ciruelos, Carmen Rubio

**Affiliations:** 1Department of Radiation Oncology, HM Hospitales, 28050 Madrid, Spain; 2Department of Medical Oncology, HM Hospitales, 28050 Madrid, Spain; 3Department of Gynecology and Obstetrics, HM Hospitales, 28050 Madrid, Spain; 4Department of Radiology, HM Hospitales, 28050 Madrid, Spain; 5Department of Medical Oncology, Hospital Sanitas La Moraleja, 28050 Madrid, Spain; 6Department of Gynecology and Obstetrics, Hospital Sanitas La Moraleja, 28050 Madrid, Spain; 7Department of Pathology, HM Hospitales, 28050 Madrid, Spain; 8Department of Nuclear Medicine, HM Hospitales, 28050 Madrid, Spain; 9Department of Plastic Surgery, HM Hospitales, 28050 Madrid, Spain

**Keywords:** chemoradiation, preoperative, neoadjuvant, TN/HER-2+ phenotype, pathological response, breast cancer

## Abstract

**Simple Summary:**

The aim of this study is to analyze the feasibility and tolerance of primary concurrent radio–chemotherapy in breast cancer patients. The primary systemic treatment of breast cancer aims to limit the scope of subsequent surgery by reducing the size of the tumor allowing for less extensive surgery, improved cosmetic outcomes, and reduced postoperative complications. We enrolled 58 patients with triple negative and HER-2 amplified breast cancer who were treated for three weeks with radiotherapy and concurrent chemotherapy. The 70.8% of patients with triple negative and the 53.1% of patients with HER-2 amplified achieved complete pathological response. In conclusion, primary concurrent radio–chemotherapy is feasible, with acceptable tolerance and high rates of pathological response.

**Abstract:**

Primary systemic treatment (PST) downsizes the tumor and improves pathological response. The aim of this study is to analyze the feasibility and tolerance of primary concurrent radio–chemotherapy (PCRT) in breast cancer patients. Patients with localized TN/HER2+ tumors were enrolled in this prospective study. Radiation was delivered concomitantly during the first 3 weeks of chemotherapy, and it was based on a 15 fractions scheme, 40.5 Gy/2.7 Gy per fraction to whole breast and nodal levels I-IV. Chemotherapy (CT) was based on Pertuzumab–Trastuzumab–Paclitaxel followed by anthracyclines in HER2+ and CBDCA-Paclitaxel followed by anthracyclines in TN breast cancers patients. A total of 58 patients were enrolled; 25 patients (43%) were TN and 33 patients HER2+ (57%). With a median follow-up of 24.2 months, 56 patients completed PCRT and surgery. A total of 35 patients (87.5%) achieved >90% loss of invasive carcinoma cells in the surgical specimen. The 70.8% and the 53.1% of patients with TN and HER-2+ subtype, respectively, achieved complete pathological response (pCR). This is the first study of concurrent neoadjuvant treatment in breast cancer in which three strategies were applied simultaneously: fractionation of RT (radiotherapy) in 15 sessions, adjustment of CT to tumor phenotype and local planning by PET. The pCR rates are encouraging.

## 1. Introduction:

Advances and improvements in the locoregional treatments of breast cancer, surgery [1,2] and radiotherapy [3,4] have contributed decisively to decreased locoregional recurrences and distant recurrences while increasing breast cancer and overall survival.

The use of primary systemic treatment (PST) in locally advanced breast cancer aims to shrink the tumor, facilitates surgery and treats possible microscopic systemic disease early. Therefore, use of PST for breast cancer patients has steadily increased in recent years [5]. This strategy is of special interest in Her-2-positive and TNBC breast cancer patients, which globally represent less than 20% of the total breast cancer diagnoses and associate worse prognosis but also are the best responders to neoadjuvant chemotherapy. Different studies have shown that reaching a complete pathological response (pCR) after PST, particularly in patients with TNBC or HER2-positive tumors, is associated with a significant survival gain; thus, achieving a high rate of pCR has become a priority objective of the neoadjuvant systemic treatment of breast cancer, especially in the most unfavorable subgroups [6,7].

In addition, radiation therapy is a mainstay of breast cancer treatment because achieving an adequate locoregional control is a cornerstone to improve the final outcome of breast cancer. About eight out of ten patients with this type of tumor are treated at some point with ionizing radiation.

Concomitant administration of radiotherapy and chemotherapy preoperatively looking for potentiating synergies is a standard approach for different tumors such as rectal, pancreatic, oesophageal or lung cancer. Concurrent radiotherapy and chemotherapy or immunotherapy regimens have been proposed prior to surgery as a strategy to improve resectability rates [8,9,10,11], increase pCR in certain breast cancers [12,13], and facilitate breast reconstruction [14]. However, concerns about its tolerance and safety have made many oncologists reluctant.

In 2018, we started a prospective pilot study of preoperative concurrent chemoradiotherapy (PCRT) in HER2+ or triple negative (TN) breast cancer patients to test feasibility and tolerance of this combination and secondarily to evaluate pathological response and oncological outcomes. Herein, we present the results from the first 58 patients enrolled in this protocol.

## 2. Material and Methods

Adult patients with proven diagnosis of HER2+ or TN non-metastatic breast carcinoma were first evaluated by a multidisciplinary breast tumor board to determine the benefit of their inclusion in this study and were offered to enroll into this prospective study. This study received approval from the Local Ethics and Clinical Committee (Code: 18.12.1241E1-GHM). Complete characteristics of the protocol have been previously described [15].

The primary study end point was to assess feasibility and tolerance of preoperative concurrent administration of chemotherapy and radiotherapy. As secondary objectives, pCR and metabolic response rates as well as locoregional control and disease-free survival rates associated to PCRT were evaluated.

After providing written informed consent, patients were enrolled to their inclusion. We included 58 localized TN/HER2+ breast cancer patients to receive PCRT. Inclusion criteria were women over 18 years old presenting with a measurable breast tumor clinically staged as cT1N+ or cT2N−/+, with adequate performance status (WHO ≤ 2), adequate bone marrow reserves (WBC count before treatment >1500/mm^3^, platelets count > 100,000/mm^3^ and haemoglobin > 10 g/L) and normal cardiac function (LVEF ≥ 50%). Every patient underwent both MRI (magnetic resonance imaging) and 18FDG-PET-CT(18 Fluorodeoxyglucose positron emission tomography) pre- and post-chemoradiation.

Patients with clinically negative nodes (cN0) first underwent a sentinel lymph-node biopsy (SLNB), while those with suspected lymph node involvement cN+ by US (Ultrasound) and/or PET required histological confirmation by using FNA (fine needle aspiration) or core needle biopsy.

Positive tumor foci detected by biopsy in the breast were marked with a coil prior to the start of treatment to facilitate their subsequent excision

Radiation treatment: All patients underwent radiotherapy simulation in an 18FDG-PET-CT with both arms raised on an immobilization device. RayStation^®^ (RaySearch Laboratories AB, Stockholm, Sweden) planning system was used for import, contouring and clinical radiation dosimetry.

Whole breast, lymph node levels I–IV and ipsilateral internal mammary chain when indicated were identified in addition to enhanced macroscopic tumor on simulation 18FDG-PET-CT. The ipsilateral and contralateral lungs, heart and the contralateral breast volume were outlined as organs at risk. The prescribed radiation dose was 40.5 Gy in 15 fractions of 2.7 Gy, five fractions per week, to whole breast and locoregional lymph node levels with simultaneous integrated boost up to 54 Gy in 15 fractions of 3.6 Gy to macroscopic tumor areas highlighted by 18FDG-PET (Figure 1a,b).

Volumetric arc radiotherapy (VMAT) was used for planning, and radiation treatment was delivered in an Elekta VERSA HDTM linac with 6 and 15 MV (Megavolts) photons and a robotic treatment couch (Elekta HexaPODTM) of 6 degrees of freedom (Elekta AB, Stockholm, Sweden). Daily repositioning was verified by VERSA X-ray kV(kilovolts)-cone-beam CT (computerized tomography) and Catalyst-HD (C-RAD, Uppsala, Sweden) systems.

### 2.1. Systemic Treatment

Concomitant chemotherapy was based on Pertuzumab–Trastuzumab–Paclitaxel followed by anthracyclines in HER2+ patients and CBDCA-Paclitaxel based regimen followed by anthracyclines in TN patients. Due to the expected risk of skin toxicity enhancement with concomitant administration of chemotherapy and radiotherapy, the sequence of chemotherapy agents was changed. So, radiotherapy was delivered concomitantly with Paclitaxel and Carboplatin, or antiHer-2 agents and Anthracyclines were administered later.

The treatment scheme is represented in Figure 2.

### 2.2. Surgery

Breast-conserving surgery was preferred when possible; however, patients with multicentric tumors at diagnosis underwent radical mastectomy irrespective of the reached response. Complete axillary lymph node dissection (cALND) was carried out in all cN+ patients regardless of their response to treatment. Those patients who underwent radical mastectomy and wished immediate reconstruction went ahead with the reconstructive process.

### 2.3. Evaluation and Statistical Analysis

Patients were assessed by the attending physician and the nurse practitioner weekly during radiation treatment, the week after and every three months afterwards. Radiation therapy induced toxicities were assessed and graded according to National Cancer Institute Common Terminology Criteria for Adverse Events version 5.0 (CTCAE 5.0). All patients underwent exhaustive cardiological assessment prior to treatment beginning and during follow-up based on an echocardiogram, lab test and periodic evaluation by a cardio-oncologist.

Clinical and radiological response was evaluated by means of an MRI and 18FDG-PET-CT at the end of planned preoperative treatment. Pathologic responses were graded according to the Miller–Payne grading system [16]. Pathologic complete response (pCR) was defined as ypT0/is ypN0.

Follow-up length was estimated from the moment of initiation of first treatment until the date of last follow-up. Locoregional free-survival (LRFS) was defined as the absence of relapse or evident progression in the irradiated volumes, both in breast/chest wall and in the treated lymph node levels. Distant metastasis-free survival (DMFS) was defined by the absence of distant metastatic tumor foci. Disease-free survival (DFS) was defined as the period free of locoregional and/or distant metastatic failure. Finally, cancer-specific survival (CSS) and overall survival (OS) were defined from the beginning of treatment until date of death from breast cancer or any other reason, respectively.

The Kaplan–Meier statistical analysis was used for the actuarial calculations using the log-rank test for the comparisons made between the survival curves, and the Pearson’s Chi square test for categorical variables was used to compare characteristics among different subgroups, and statistical significance was considered when reaching a *p* value < 0.05. The SPSS (IBM SPSS Statistics for Windows, Version 19.0. (Armonk, NY, USA: IBM Corp.)) package software was used for all calculations.

## 3. Results

### 3.1. Patients’ Characteristics

From October 2018 to July 2021, 58 female patients were enrolled. A total of 56 patients with a median follow-up of 22.2 months (range 4.1–49) completed primary chemoradiation and surgery.

Patients’ characteristics are summarized in Table 1. With a median age of 52 years-old (range 31–78), most of them were T2 (43/58 = 74%) according to AJCC staging. A total of 43% of them had TN phenotype, while the remaining 57% were defined as HER2+ (38% hormone receptor (HR) positive and 19% HR negative).

A total of 37 patients underwent conservative breast surgery, while a modified radical mastectomy was performed on 19 patients. Axillary lymphadenectomy was practiced on 29 patients. Most of those who underwent mastectomy chose to carry out reconstruction. Since many patients are still in the reconstruction process, a specific assessment on this matter will be the reason for a future sub-analysis.

### 3.2. Treatment Tolerance and Adverse Effects

Acute skin toxicity was the most frequent adverse effect of the combined treatment. A total of 77.6% of patients (*n* = 45) had G (grade) 1 dermatitis, 13.8% (*n* = 8) G2 and 8.6% G3 (*n* = 5). Some 39.6% of patients (*n* = 23) required a partial reduction of chemotherapy dose or the suspension of any of the cycles due to adverse events. A total of 43% of patients (*n* = 25) experienced G2 cytopenia, 26% G3, and a total of eight patients (13.7%) suffered from G4 cytopenia. Of the 58 patients, 2 were lost for follow-up before finishing chemoradiation. The first patient died due to a generalized sepsis after chemotherapy-induced pancytopenia at the age of 65. The second patient died of causes unrelated to treatment at the age of 78.

Four patients (7%) experienced a transient drop of more than 10 points in left ventricle ejection fraction (LVEF), although all patients returned to normal cardiac function after completion of treatment.

Fifty-six patients completed upfront chemoradiotherapy undergoing definitive surgery. A total of nine patients (16%) developed postoperative complications: infection after implanting a breast tissue expander in three patients, an axillary postsurgical seroma in one patient and persistent postsurgical breast seroma in five patients. All cases resolved after antibiotic therapy. A complete description of the observed toxicities is detailed in Table 2.

### 3.3. Response Analysis

A total of 56 patients (96.5%) completed the planned chemoradiation treatment; 60.7% (*n* = 34) reached a complete pathological response and most of them (*n*: 49; 87.5%) achieved at least >90% loss of tumor cells. Pathologic response rates are detailed in Table 3.

All patients underwent an MRI and 18FDG-PET-CT before chemoradiation treatment. However, six patients were not evaluated with 18FDG-PET-CT after completing PCRT and before surgery. The metabolic response assessed by 18FDG-PET-CT was considered complete in 50 patients (87.9%) and partial in 2 patients. Radiological response measured by MRI was considered complete in 41 patients (70.7%) and partial in 14 patients (24.1%). The calculated negative predictive value (NPV) of 18FDG-PET-CT and MRI were 61.2% and 73.1%, respectively.

Grade 3 tumors, Ki-67 index above 42% and TN and HER2+ HH- tumors achieved.

higher rates of pCR (*p* < 0.05 for all comparisons). Complete univariate analysis for pCR is detailed in Table 4.

### 3.4. Disease-Free Survival and Overall Survival

To date, the 100% of patients who completed PCRT are free of local relapse and alive. Median follow-up of this study is 24.2 months (range 4.1–49.4 months). One patient with a TN right-sided breast cancer developed a contralateral tumor after 25.3 months of follow-up, but it was histologically different from the primary one, being a cT1 luminal A breast cancer. This patient underwent BCS followed by radiation therapy and hormone therapy, being free of disease on last follow-up.

## 4. Discussion

Survival among women with breast cancer has been found to be closely related with the immunohistochemical profile. Those tumors with HER2 gene overexpression or TN had lower survival rates, although extended use of primary systemic treatment (PST) with dual HER2 blockade in HER2+ patients and cisplatin-based chemo and immunotherapy in TN patients have dramatically changed the current landscape for both aggressive subtypes. Reaching a complete pathological response after PST in TN or HER2+ tumors is associated with a significant survival gain [6,7,17,18] and is now considered a surrogate marker for it.

For TN breast cancer, the use of conventional chemotherapy regimens based on the combination of anthracyclines, paclitaxel, and cyclophosphamide as part of the PST, was associated with pCR rates of 35–45% [19]. Addition of platinum compounds to PST schemes showed an increase in pCR rates. A recent meta-analysis of nine randomized studies including more than 2000 women with TN tumors showed that platinum-based PST increased the pCR rate from 37% to 52.1% (*p* < 0.001) [20]. Advances in modern immunotherapy are contributing to definitively change the therapeutical approach, and several studies are currently investigating the addition of immunotherapy to standard chemotherapy for TN tumors [21]. Combining these new drugs to enhance the immunogenicity of TN breast cancer together with the ability of radiotherapy to modulate the different stages of the immune response against cancer, from the generation and release of tumor antigens to the possibility of inducing cell death mediated by T lymphocytes, will be a promising alternative for these women [16].

Among HER2+ patients, development of specific drugs against the HER2 receptor completely changed their prognosis, and now these targeted therapies are considered part of the standard treatment in these women [22]. In the neoadjuvant setting, the addition of trastuzumab showed a significant increase in the probability of achieving pCR (from 21% to 38%; *p* < 0.001), becoming a mainstay of the PST [23]. However, the persistence of relapses, metastasis, or resistance to trastuzumab has prompted the development of other drugs targeting HER2. A recent meta-analysis of nine randomized studies (*N* = 2758) reported that double blockage by co-administration of trastuzumab and pertuzumab/lapatinib/neratinib significantly increased the probability of reaching pCR [24].

The use of radiotherapy prior to surgery is a frequent alternative in many tumors (i.e., rectum, sarcomas, pancreas, etc.) and is associated with good pathological response rates that facilitate surgery and improve prognosis of cancer patients. In addition, in breast cancer, clinical studies with long follow-up have shown that exclusive preoperative radiotherapy achieves rates of pCR in 8–11% of patients [8,25,26,27,28]. Furthermore, an analysis of the SEER (Surveillance, Epidemiology, and End Results) including more than 250,000 women concluded that preoperative radiotherapy would be associated with an increase in disease-free survival [12]. Although concurrent delivery of chemotherapy and radiotherapy is practiced in other tumor locations showing an increase not only in local control but also in survival rates, this combination has not been widely used in breast cancer patients. Concomitant radiotherapy with radio sensitizing chemotherapy in breast cancer has been tested mostly in metastatic, recurrent or inoperable or inflammatory breast cancer patients [29,30,31]. Recent advances in neoadjuvant treatments have renewed the interest in exploring the combination of chemotherapy and radiation therapy in breast cancer, especially in the most aggressive and unfavorable molecular subtypes. Table 5 summarizes published outcomes from concurrent neoadjuvant chemoradiotherapy regimens followed by surgery.

Despite the promising pCR rates achieved in different trials of preoperative chemoradiation therapy for breast cancer, many oncologists and surgeons are still reluctant to use it because of tolerance concerns during treatment or the risk of complications at the time of definitive surgery. In our experience, the toxicity profile during treatment was acceptable. The presence of mild (76% G1) or moderate (12% G2) radio-induced skin toxicity was the most frequently observed acute complication. In addition, there were 13 patients (31%) with cytopenia G3 and 4 patients (9%) with cytopenia G4. One of them died due to a generalized sepsis after chemo-pancytopenia at the age of 65. All patients completing planned treatment underwent surgery according to what was established before starting preoperative chemoradiotherapy. Although there were some controversies with respect to the influence of upfront concomitant chemoradiotherapy on ulterior surgery, the 16% rate of postoperative complications in our series is like those described in other studies.

Pathologic complete response rates of preoperative combined regimens vary from 9–45%. In our series, the pCR rates of patients with TN and HER2+ tumors were 70.8% and 53.1%, respectively, slightly higher than those previously reported. Different factors could contribute to explain these differences: (1) all patients treated with our protocol had HER2+ or TN tumors, while the percentage of both reported by other authors is highly variable but lower than in our series; (2) all previous studies used radiation therapy schedules with conventional fractionation for breast, regional nodes and boost irradiation, while our series employed an accelerated hypofractionated scheme with simultaneous integrated boost (SIB) that allowed us to slightly increase the final dose in defined areas of greatest tumor burden according to the simulation 18FDG-PET-CT; and (3) the concurrent chemotherapy regimen in our series was specifically chosen for each molecular subtype, unlike the more uniform regimens used by other studies. Not surprisingly, observed pCR rates in our series were higher for TN than for HER2+ tumors. However, when the HER2+ component is separately analyzed between pure HER2+/HR− tumors and HER2+/HR+ (luminal B) tumors, a difference is observed in the pCR rates between both, being 63.6% and 47.6%, respectively. The lower probability of achieving a pCR with preoperative treatment in luminal B tumors has already been demonstrated with respect to neoadjuvant chemotherapy in the NeoSphere trial [44] and in relation to neoadjuvant chemoradiation by Adams et al. [37].

We are conscious pf several weaknesses and strengths of our study. The small sample size is one of the most important limitations since it reduces the statistical power after any stratification. Likewise, the inclusion of patients in this study was not consecutive since the evaluation of each patient was submitted to a multidisciplinary board, so there may also be a certain selection bias. Finally, the follow-up is still very short, so it is not possible to analyze the impact of the pathological response on survival nor to evaluate the long-term cosmetic result yet. On the other hand, the feasibility, safety, and acceptable tolerance of this moderate hypofractionated radiotherapy with a simulation technique based on 18FDG-PET-CT for a more precise definition of tumor harboring areas together with concurrent administration of a tailored chemotherapy regimen adjusted to the molecular subtype in a neoadjuvant setting could contribute to optimizing the treatment sequence by shortening its total duration while allowing encouraging pCR rates, especially in the triple negative subtypes. In parallel, preoperative administration of chemotherapy and radiotherapy would facilitate both oncoplastic surgery in the case of a conservative approach as well as the immediate breast reconstruction techniques.

## 5. Conclusions

Individualization of treatment approaches is an exciting challenge in breast cancer, especially due to the heterogeneity and variety of cancer subtypes. Combining preoperative highly conformed radiation techniques with tailored systemic therapies adjusted to molecular subtypes (TN and HER-2+) seems feasible and well tolerated achieving, in addition, non-negligible tumor response rates. According to existing evidence, the impact of the pCR on tumor control and survival is higher in HER2+/TN tumors, being adopted as surrogate end point. Thus, rate of pCR observed in this pilot study, especially in patients with triple negative tumors, opens the possibility to future prospective and multicenter randomized trials to establish the definitive role of preoperative concurrent chemoradiation therapy in selected breast cancer patients.

## Figures and Tables

**Figure 1 cancers-14-04531-f001:**
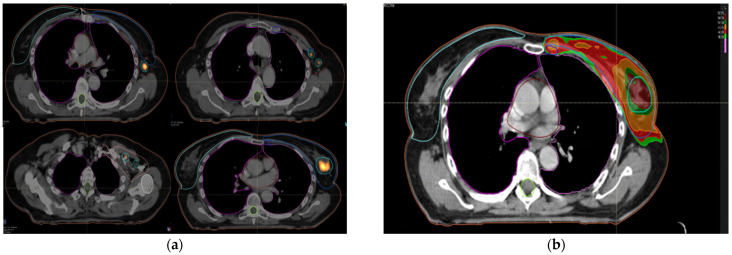
(**a**) contours on planning 18FDG-PET-CT; (**b**) dosimetric planning with VMAT.

**Figure 2 cancers-14-04531-f002:**
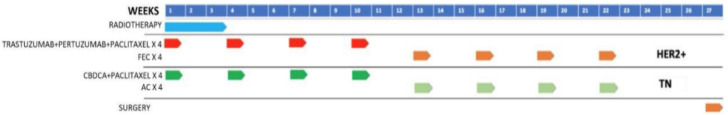
Chemoradiation treatment scheme.

**Table 1 cancers-14-04531-t001:** Patients’ characteristics.

			*N*	Frequency
Number of patients		58		
Age (median)		52 (31–78)		
Breast side		Right side	35	60%
		Left side	23	40%
Histology		Ductal infiltrating carcinoma	58	100%
Grade		G2	24	41%
		G3	34	59%
Tumor size (median)		28 mm (16–90)		
		Staging at diagnosis		
T		T1	2	3%
		T2	43	74%
		T3	12	21%
		T4	1	2%
N		cN+	26	45%
		cN0 (SNLB negative)	28	48%
		cNX (no migration)	4	7%
Molecular subtype		HR+/HER2+ HR−/HER2+ HR−/HER2−	22 11 25	38% 19% 43%
Ki 67 index		Median 43%; average 47%		
		<20% >20%	5 53	9% 91%

SLNB: sentinel lymph node biopsy; TILS: tumor infiltrating lymphocytes; HR: hormone receptor; NA: not available.

**Table 2 cancers-14-04531-t002:** Treatment tolerance according to CTCAE scale.

	G0-1-2		G3		G4	
	*n*	%	*n*	%	*n*	%
Dermitis	53	91.4	5	8.6	0	0.0
Cytopenia	36	62.1	15	25.9	7	12.1
Alopecia	58	100	0	0.0	0	0.0
Nauseas	57	98.3	1	1.7	0	0.0
Mucositis	57	98.3	1	1.7	0	0.0
Diarrhea	54	93	3	5.2	1	1.7
Neurotoxicity	58	100	0	0.0	0	0.0

**Table 3 cancers-14-04531-t003:** Pathologic response rates (graded by the Miller–Payne classification).

			G1-2		G3		G4		G5	
	Total	%	*n*	%	*n*	%	*n*	%	*n*	%
TN	25	43	0	0	4	16.7	3	12.5	17	70.8
HER-2+	33	57	0	0	3	9.4	12	37.5	17	53.1
HER-2+ HH+	22	38	0	0	2	9.5	9	42.9	10	47.6
HER-2+ HH−	11	19	0	0	1	9.1	3	27.3	7	63.6
Total	58	100	0	0	7	12.5	15	26.8	34	60.7

**Table 4 cancers-14-04531-t004:** Univariate analysis of factors influencing pCR.

Pathological Complete Response(pCR)		Pearson’s Chi-Square p
Age	<52 vs. ≥52	0.06
Grade	G3 vs. G2	0.02
Size	<28 mm vs. ≥28 mm	0.09
ki 67	<43 vs. ≥43	0.01
Phenotype	TN vs. Her2+	0.03
Phenotype	Her2+ HH+ vs. Her2+ HH−	0.05
PET (SUV value)	<7.2 vs. ≥7.2	0.07

**Table 5 cancers-14-04531-t005:** Summary of studies of concurrent chemoradiation in breast cancer.

Author	*N*	Type of Study	HER2+/TNBC	Preoperative Radiotherapy	Preoperative Systemic Treatment	pCR (%)	Acute Cutaneous Toxicity	Post-Surgical Complications
Semiglazov 1994 [32]	271	Randomized neoadjuvant chemoradiotherapy (CTRT) vs. neoadjuvant radiotherapy (RT)	HER2+: NS TNBC: NS	WBI: 60 Gy (2 Gy) RNI: 40 Gy (2 Gy)	Concurrent: TMF vs. No CT	pCR 29.1% (CTRT) vs. pCR19.4% (RT) *p* < 0.05	G1-2: 6.5% (CTRT) vs. 8.9% (RT)	22.5%
Formenti 1997 [33]	35	Prospective	HER2+: 82.8% TNBC: NS	WBI+RNI 50 Gy (2 Gy)	Pre-RT: 5-FU → Concurrent: 5-FU	pCR: 20%	G2: 26%	NR
Skinner 2000 [34]	28	Prospective	NS	WBI + RNI 45 Gy (1.8 Gy)	Concurrent: paclitaxel	pCR 26%	G2: 30%	41%
Formenti 2003 [35]	44	Prospective	HER2+: NS TNBC: NS	WBI + RNI 45–46 Gy (1.8–2 Gy)	Pre-RT: docetaxel → Concurrent: docetaxel	pCR 34%	G2: 45% G3: 6.8%	14%
Chakravarthy 2006 [36]	38	Prospective	HER2+: 34% TNBC: NS	WBI + RNI 45 Gy (1.8 Gy)	Pre-RT: Paclitaxel → Concurrent: Paclitaxel	pCR 34%	G3: 2.6% G4: 2–6%	NR
Bollet 2006 [37] 2012 [28]	60	Prospective	HER2+: 14% TNBC: 20%	WBI: 50 Gy (2 Gy) ± boost 10 Gy RNI: 46 Gy (2 Gy)	Concurrent: Vinorelbine + 5-FU	pCR 27%	G2: 19% G3: 14%	36.6%
Gaui 2007 [38]	28	Retrospective	HER2+: NS TNBC: NS	WBI + RNI 50 Gy (2 Gy)	Concurrent: Capecitabine	pCR 9%	G1: 35% G2: 11%	3.6%
Shanta 2008 [39]	1117	Retrospective	HER2+: NS TNBC: NS	WBI + RNI 40 Gy (2 Gy)	Concurrent: CMF/ECF/FAC	pCR 45.1%	NR	20.8%
Alvarado- Miranda 2009 [40]	112	Retrospective	HER2+: 1.7% TNBC: 60%	WBI + RNI 50 Gy (2 Gy) + boost 10 Gy	Concurrent: Mytomicin + 5Fu or Gemcitabine + cDDP	pCR29.5%	G3: 22.4%	NR
Adams 2010 [41]	105	Pooled analysis from 3 prospective trials, including [28] and [Formenti2003]	HER2+: 32% TNBC: 22.8%	WBI + RNI 45 Gy (1.8 Gy) ± boost 14 Gy	Concurrent:Paclitaxel± Trastuzumab	pCR 23%	NR	NR
Monrigal 2011 [42]	210	Retrospective	HER2+: 9% TNBC: NS	WBI + RNI 50 Gy (2 Gy) + boost 10 Gy	Concurrent: Anthracyclin based CT ± Paclitaxel ± Trastuzumab	pCR 35.2%	NR	21.6%
Matuscheck 2012 [43]	315	Retrospective	HER2+: NS TNBC: NS	WBI + RNI 50 Gy (2 Gy) + boost 10 Gy	EC/CMF/AC/Mitoxantrone Pre-RT: 61% or Concurrent: 36% or No CHT: 3%	pCR 29.2%	NR	NR
Brackstone 2017 [9]	32	Prospective	HER2+: 11.1% TNBC: 18.5%	WBI + RNI 45 Gy (1.8 Gy) ± boost 5.4 Gy	Pre-RT: FEC → Concurrent: Docetaxel	pCR 22.6%	G3: 25%	3%
Current series 2022	58	Prospective	HER2+: 57% TNBC: 43%	WBI + RNI 40.5 Gy (2.7 Gy) + SIB 54 Gy (3.6 Gy)	Concurrent: Pertuzumab–Trastuzumab–Paclitaxel → AC in HER2+ Concurrent: CBDCA-Paclitaxel → AC in TNBC	TN: 71% HER2+ 53% HR+: 48% HR−: 64%	G1: 78% G2: 14% G3: 5%	16%

pCR: pathologic complete response; WBI: whole breast irradiation; RNI: regional node irradiation; TMF: thiotepa-metrotexate-5 fluorouracil; 5-FU: 5 fluorouracil; EC: epirrubicin-cyclophosphamide; AC: adryamicin-cyclophosphamide; FEC: 5 fluorouracil-epirrubicin-cyclophosphamide; FAC: 5 fluorouracil-adryamicin-cyclophosphamide; cDDP: cisplatin; CBDCA: carboplatin: NS: not specified; NR: not reported.

## Data Availability

Data generated or analyzed during the study are available from the corresponding author by request.

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
