# Peer review of "Primary Chemoradiotherapy Treatment (PCRT) for HER2+ and Triple Negative Breast Cancer Patients: A Feasible Combination"

_cancers, 2022, doi:10.3390/cancers14184531_

Round 1

Reviewer 1 Report

I appreciate the opportunity to review this manuscript and learn more about this research. Overall, the article is a fascinating topic. The findings are fantastic and are presented by the authors in an impeccable way. The data are excellent as well. But the introduction section should have relevance and a theoretical basis, providing readers with sufficient information about previous research results to follow current research's basic principles and procedures. So, I think authors should add literature support regarding studies.  The manuscript can be accepted with some improvements. 

Author Response

We really appreciate your comments. We agree there is more evidence to support and explain the rationale of this approach, hence we have added so more data. Please see the attachment with the " track changes" word document:  

Reviewer 2 Report

Dear Authors:

The manuscript "Primary chemoradiotherapy treatment (PCRT) for HER2+ and Triple Negative breast cancer patients: A feasible combination." by Ciérvide et al has demonstrated primary concurrent radio-chemotherapy is feasible, with acceptable tolerance and high rates of pathological response. I have just afew suggestions.

1. Some background information or reference is missing: Please add more background information about breast cancer, especially the surgery of breast cancer. (please cite:

1. Efficacy of da Vinci robot-assisted lymph node surgery than conventional axillary lymph node dissection in breast cancer - A comparative study. Int J Med Robot. 2021 Dec;17(6):e2307. doi: 10.1002/rcs.2307. Epub 2021 Jul 29. PMID: 34270843.

2. Robot-Assisted Minimally Invasive Breast Surgery: Recent Evidence with Comparative Clinical Outcomes. J Clin Med. 2022 Mar 25;11(7):1827. doi: 10.3390/jcm11071827. PMID: 35407434; PMCID: PMC8999956.)

2. The manuscript needs linguistic improvement.

Best,

Author Response

Thank you very much for your suggestions. 

We have increased the background information empowering the advances and improvements in the locoregional treatments as surgery (2 suggested cites) and radiotherapy.

English grammar and style have been reviewed.

Please see the "attached" tracked changes word document.

Reviewer 3 Report

It was a manuscript about the clinical evaluation of chemoradiotherapy for two types of breast cancer (HER 2+ and triple-negative). Here are some comments on this study that should be considered before publication: 

1-      There are some typos- and grammatical mistakes in the text that should be corrected.

·       spective study. of. This study

·       patient and persistent post-surgical breast seroma in 5 patients. All cases resolved after

...

2-      G1 dermatitis, 13.8 % (n=8) G2 and 8.6 % G3” what do G1, G2, ... refer to? Please write all the abbreviations on their first-time usage.

3-      Please rewrite the caption of table 2.

Author Response

I am very grateful to your comments.

--Abbreviations have been explained in the text:

--CT (chemotherapy) RT (radiotherapy), MRI (magnetic resonance imaging),  18FDG-PET-CT (18 Fluorodeoxyglucose positron emission tomography), US (Ultrasound), MV (Megavolts), KV (Kilovolts), CT (Computerized tomography), G (grade)

--Mistakes have been solved:

-In “Material and Methods” section:

“…. to determine the benefit of their inclusion in this study and secondary were offered to ENROL into this prospective study.  of.” has been replaced by “…. to determine the benefit of their inclusion in this study and secondary were offered to ENROLL  into this prospective study”.

-In “Results: 3.2. Treatment tolerance and adverse effects” section:

“All cases resolved after.” has been replaced by “ All cases resolved after antibiotic therapy. Complete description of the observed toxicities is detailed in Table 2.

The caption of table 2 “ Tolerance” has been replaced by: Treatment tolerance according to CTCAE scale

--In “Results: 3.2. Treatment tolerance and adverse effects” section:

 G has been replaced by Grade 

Please see the "attachment" tracked changes word document
